

# Impact of proximal cavities and primary molar absence on space in the dental arches

Rafael T. Gomide[1], Jo E. Frencken[2], Soraya C. Leal[1], Anne Marie Kuijpers-Jagtman[3,4,5] and Jorge Faber[6]

[1] Division of Pediatric Dentistry, Dental School, Universidade de Brasília, Brasília, Distrito Federal, Brazil
[2] Department of Dentistry, Section of Function and Prosthetic Dentistry, Radboud University Medical Centre, Nijmegen, Netherlands
[3] Department of Orthodontics, University Medical Center Groningen, Groningen, Netherlands
[4] Department of Orthodontics and Dentofacial Orthopedics, University of Bern, Bern, Switzerland
[5] Faculty of Dentistry, Universitas Indonesia, Jakarta, Indonesia
[6] Unaffiliated, Brasília, Distrito Federal, Brasil

Corresponding author
Rafael T. Gomide,
rafael_gomide@hotmail.com

## ABSTRACT

**Background:** A recently proposed treatment protocol for dental caries in primary teeth, called Ultra Conservative Treatment (UCT), keeps medium to large cavities open so that children can keep them clean with tooth brushing and fluoride toothpaste. However, carious lesions have been related to malocclusion and decrease of space for the eruption of the permanent successor.
**Methods:** This cross-sectional study evaluated dental casts of 235 schoolchildren, aged 6–7 years old of six public schools in deprived suburban areas, and with at least two cavitated dentin carious lesions. The casts were grouped according to the location of cavitated dentin lesions into non-proximal cavity (NPC), proximal cavity with buccal or lingual surface contact point to adjacent tooth (PCCP) and proximal cavity without contact to adjacent tooth (PCWC), as well as the absence of primary molars due carious lesions (PMA). The relationship between location of cavitated dentin lesions or absence of primary molars with C+D+E and D+E space was analyzed.
**Results:** Children with absence of primary molars exhibited the smallest C+D+E and D+E space in the maxilla ($P < 0.001$) and mandible ($P < 0.001$), followed by proximal cavity without buccal or lingual surface contact. No significant difference was observed between NPC and PCCP groups in upper and lower arches.
**Discussion:** PCWC are associated with minor (less than the leeway space) C+D+E and D+E space loss in both arches, and additional space loss is noted when primary molars are prematurely lost. These results may have implications for orthodontic and restorative dental care decisions in children.

## INTRODUCTION

Efforts to improve global oral health have led to the development of new materials, techniques and oral health protocols for use in communities that suffer from limited access to oral health services (*Marcenes et al., 2013*; *Peres et al., 2010*; *Yee & Sheiham, 2002*). Ultra Conservative Treatment (UCT) is one such new protocol and it implies restoring small cavities in primary teeth according to the Atraumatic Restorative Treatment (ART) method and cleaning medium-and-large-size cavities plaque-free with toothbrush and fluoride toothpaste. The effectiveness of UCT has been compared to ART and a conventional restorative treatment (CRT; amalgam) protocol in a clinical trial. Although the trial found the 3.5-year cumulative survival percentage of UCT-treated primary molars not to be significantly different from the survival percentage of cavitated teeth treated according to the ART and CRT protocol (*Mijan et al., 2015*), UCT-treated second primary molars exfoliated earlier than ART and CRT restored primary molars (*Mijan et al., 2014*). Thus, UCT may be a suitable treatment protocol from a cariological and restorative point of view but it is uncertain whether medium-and-large-size cavities—and even primary molars' early loss—may impact on the eruption of premolars and/or affect normal development of the dentition and hence may have orthodontic implications in the long-term.

The impact of untreated cavitated dentin carious lesions in primary teeth on the development of the permanent dentition has received relatively little attention, and results are inconsistent. For instance, results of cross-sectional studies vary from an association between dmft/DMFT index and malocclusion (*Gábris, Márton & Madlénaa, 2006*; *Mtaya, Brudvik & Astrøm, 2009*; *Nalcaci et al., 2012*) to no relationship at all (*Borzabadi-Farahani, Eslamipour & Asgari, 2011*). One study even identified children with cavitated primary teeth as being less prone to presenting malocclusion in the primary dentition than those that had sound primary teeth (*Stahl & Grabowski, 2004*), whereas a 6-year longitudinal study suggested that large cavities tend to decrease the space for the erupting premolars (*Northway, Wainright & Demirjian, 1984*). These contrasting results are partly explained by the different research methods used and the composition of the populations in which the studies were performed.

The status of the primary molars may affect the space conditions in the buccal segments of the dental arch during the transitional period. An appropriate measurement to assess the space conditions for the erupting premolars is size of the D+E space (*Northway & Wainright, 1980*). Inclusion of a measurement of the space for the erupting canine (C+D+E space) provides further insight into the space conditions in the buccal segments of the dental arch. This could be especially important in the upper arch where the eruption of the premolars occurs before the eruption of the canines.

The fact is that there is a tremendous imbalance between the number of studies that address how to treat malocclusion and those that focus on understanding its etiology. We used data from a deprived community under treatment with the UCT protocol to investigate the possible relationship between presence of primary molars with cavitated dentin carious lesions and/or absence of primary molars due to caries and the available

C+D+E and D+E spaces. In the current study, we tested the null hypothesis that there is no difference in C+D+E and D+E spaces in schoolchildren in relation to the status of the proximal surface of posterior primary molars.

## MATERIALS AND METHODS

### Study design

This study was approved by the Research Ethics Committee of the University of Brasília Medical School (protocol 081/2008) and was registered at the Dutch Trial Registration Centre (protocol 1699). In this cross-sectional study we used baseline data from a clinical trial that compared three restorative treatment protocols (*Mijan et al., 2014*) which have been described in detail elsewhere (*Mijan et al., 2014*). A brief description is presented below.

All children attending six public primary schools of a socially deprived suburban area of Brasilia, Brazil, were examined epidemiologically in April-May on 2009 (*De Amorim et al., 2012*). Only children with good general health, at least two cavitated dentin carious lesions in primary molars without pain and pulp involvement and whose parents/guardians signed the informed consent were considered eligible to participate. A total of 302 6-and 7-year olds were included in the clinical trial that compared the three restorative treatment protocols (*Mijan et al., 2014*, *2015*).

Children were allocated to three treatment protocol groups: CRT as the control group and ART and UCT as the test groups. The unit of sampling was the school. As only two of the six schools had a dental unit with rotary equipment, these schools constituted the CRT group. The remaining four schools were randomly allocated to ART and UCT groups by the flip of a coin. No effect was observed regarding gender ($p = 0.71$) and mean dmft-score ($p = 0.75$) between the three treatment groups at baseline (*Mijan et al., 2014*).

### Dental cast analysis

Immediately after completion of the restorative treatment protocol, impressions of both dental arches and a wax bite were taken using full autoclavable mouth trays and alginate (Avagel, Dentsply, Petrópolis, Brasil). The impressions were poured in plaster within 1 h.

Occlusal photographs of the casts were taken with an SLR camera (D40, Nikon, Japan) equipped with a 105 mm Sigma Macro zoom lens (model EX DG Macro; Sigma–Aldrich, St. Louis, MO, USA). A copy stand with a clear glass top was built so that the camera lens faced up and its long axis remained perpendicular to the glass. Photographs were taken with the casts facing down, the occlusal plane over the glass. A ruler was placed beside each model and framed in the photograph; it was later used to calibrate the morphometric program. All photographs were taken with standardized lighting and focal distance.

### Intra-arch variables

Measurements for these variables were performed on the occlusal pictures of both dental arches using a morphometric program (Digimizer v.4.2; MedCalc Software, Belgium). All measurements were performed by one calibrated examiner (RG) at the same time

(between 8 and 10 am) every day for about 6 weeks at the same location under the same lighting conditions. The following variables were measured:

D+E space: Distance from the most mesial point of the first permanent molar to the most distal point of the primary canine in both arches (in mm) on both sides. Whenever the first permanent molar was absent, the most distal point of the second primary molar or premolar was considered. If the primary canine was absent, the most mesial point of the first primary molar or premolar was considered.

C+D+E space: calculated as the sum of the D+E space and the distance between the most distal point of the primary canine to the most distal point of the lateral primary or permanent incisor. Whenever the primary canine was absent, the most mesial point of the first primary molar or premolar was considered. If the primary or permanent lateral incisor was absent, the most mesial point of the primary canine was considered. Measurements were performed in both arches.

## Primary molar status

Presence of cavitated carious lesions at the mesial and distal surfaces of primary molars was assessed from the occlusal photographs of the casts presented on a tablet with magnification between 2 and 8 times (Samsung Galaxy Note 10.1, Suwon, South Korea) by two calibrated evaluators independently (JF, RG). Differences were discussed until consensus was reached. Absent primary molars were considered prematurely lost due to caries. All primary molars were grouped according to the possible impact of the condition on tooth migration (Fig. 1): Non-Proximal Cavity (NPC, which included occlusal, buccal and/or lingual cavitated dentin carious lesions and sound teeth), Proximal Cavity with buccal or lingual surface Contact Point to adjacent tooth (PCCP), Proximal Cavity Without Contact to adjacent tooth (PCWC) and Primary Molar Absent (PMA). Each quadrant was assessed independently and scored according to the most severe feature.

## Statistical analyses

Sample size had been calculated for the controlled clinical trial, which aimed to evaluate the survival rate of primary molars using three restorative treatment protocols (Mijan et al., 2015). In brief, sample size was set at 88 individuals per group ($\alpha = 0.05$; $1-\beta = 0.8$), including a 10% correction for dependency on treatments within a child, and an 8% estimated annual loss of children (Mijan et al., 2015).

Intra-examiner reliability for the intra-arch variables was calculated using paired sample correlation, in which 10% of the baseline measurements were reassessed whereas inter-examiner agreement for primary molar status was calculated as Cohen's kappa coefficient. A kappa < 0 reflects "poor", 0–0.20 "slight", 0.21–0.40 "fair", 0.41–0.60 "moderate", 0.61–0.80 "substantial" and from 0.81 "almost perfect" agreement.

Descriptive statistics related to age and gender were obtained for each group. A flowchart of patient and model allocations as well as statistical analyses is presented in Fig. 2. A chi-square test was used to access possible gender unbalance, and $t$ tests were used to assess whether there were differences between right and left sides. ANOVAs were applied to assess the impact of the independent variable "primary molar's status"

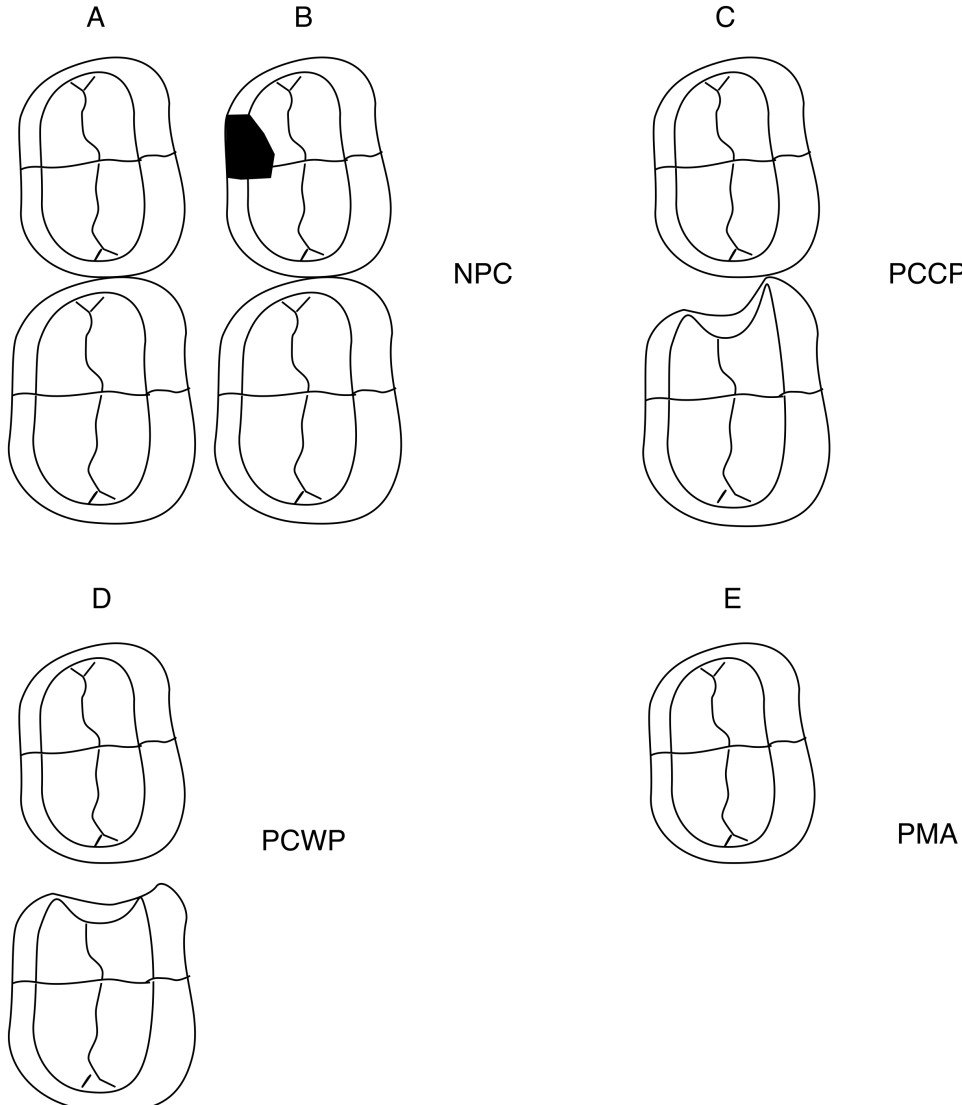

**Figure 1 Ilustration to describe different primary molars carious conditions and the groups of the primary molar status.** (A) Primary molar with sound proximal surface. (B) Primary molar with a carious lesion in a non-proximal surface. (C) Primary molar with an open proximal cavity of which the proximal-buccal or proximal-lingual surface is in contact with the adjacent tooth. (D) Primary molar with an open proximal cavity without a proximal-buccal or proximal-lingual surface contact with the adjacent tooth. (E) Primary molar absent or extracted because of pulp exposure, ulceration, fistula or abscess.

(NPC, PCCP, PCWC and PMA) on the dependent variables (C+D+E space and D+E space). Whenever statistically significant differences were found, post hoc Tukey's tests were performed. The alpha level was set at 5%.

## RESULTS

### Disposition of participants

From the original 302 schoolchildren included in the clinical trial, 25 children were excluded from the current study upon re-diagnosing the condition of the cavity as having

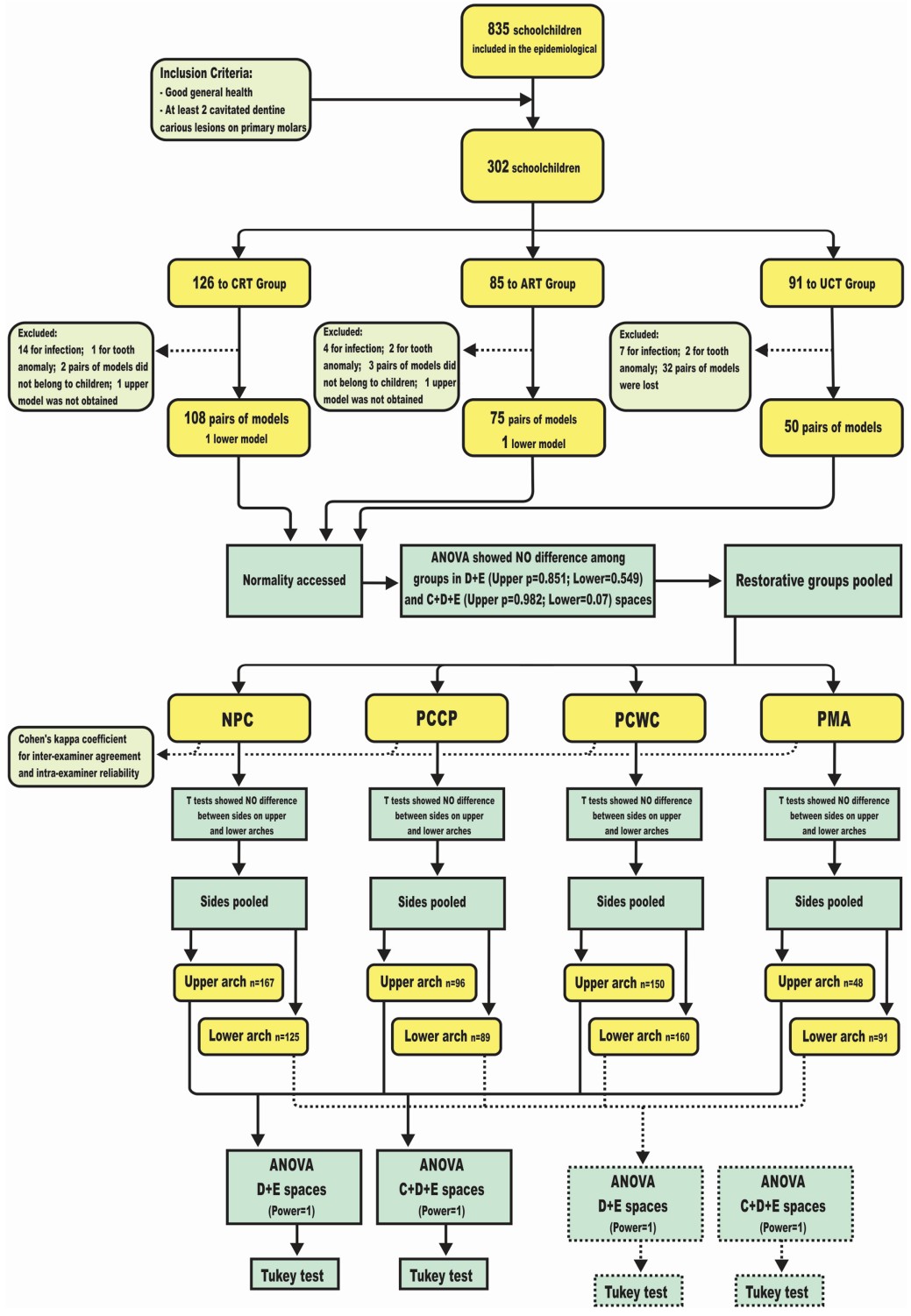

**Figure 2 Flowchart demonstrates patient and study model allocations, as well as the statistical methods.** CRT, conventional restorative group (Amalgam); ART, atraumatic restorative group; UCT, ultra conservative group; NPC, non proximal cavity; PCCP, proximal cavity with buccal or lingual surface contact point to adjacent tooth; PCWC, proximal cavity without contact point to adjacent tooth; PMA, primary molar absent.

**Table 1 Intra-examiner reliability, mean difference between measurements (in mms) and their 95% confidence interval for maxilla and mandible C+D+E and D+E spaces.**

| Arch | Space | r | P | Mean difference | 95% CI |
|------|-------|-----|--------|-----------------|--------|
| Maxilla | C+D+E | 0.99 | <0.001 | 0.03 | [−0.04 to 0.09] |
| | D+E | 0.99 | <0.001 | 0.01 | [−0.04 to 0.05] |
| Mandible | C+D+E | 0.99 | <0.001 | 0.05 | [−0.02 to 0.12] |
| | D+E | 0.99 | <0.001 | <0.01 | [−0.04 to 0.05] |

**Table 2 Comparison of right and left side of the C+D+E and D+E space (mm) according to primary molars status.**

| | Distance | Maxilla | | | | | | Mandible | | | | | |
|---|----------|---------|---|------|---|---|---|----------|---|------|---|---|---|
| | | Right | | Left | | | P | Right | | Left | | | P |
| | | Mean ± SD | N | Mean ± SD | N | | | Mean ± SD | N | Mean ± SD | N | | |
| NPC | D+E space | 16.93 ± 0.97 | 65 | 16.65 ± 1.04 | 101 | 0.090 | | 18.40 ± 0.92 | 63 | 18.30 ± 1.07 | 62 | 0.581 | |
| | C+D+E space | 24.05 ± 1.26 | 65 | 23.80 ± 1.39 | 101 | 0.245 | | 24.47 ± 1.22 | 63 | 24.21 ± 1.52 | 62 | 0.307 | |
| PCCP | D+E space | 16.67 ± 0.97 | 55 | 16.76 ± 0.97 | 41 | 0.640 | | 18.04 ± 0.97 | 46 | 18.15 ± 0.89 | 43 | 0.576 | |
| | C+D+E space | 23.97 ± 1.37 | 55 | 24.15 ± 1.31 | 41 | 0.519 | | 24.12 ± 1.38 | 46 | 24.08 ± 1.34 | 42 | 0.890 | |
| PCWC | D+E space | 16.39 ± 1.06 | 85 | 16.27 ± 0.91 | 64 | 0.456 | | 17.54 ± 1.20 | 77 | 17.49 ± 1.13 | 83 | 0.807 | |
| | C+D+E space | 23.45 ± 1.55 | 85 | 23.36 ± 1.33 | 64 | 0.698 | | 23.65 ± 1.49 | 77 | 23.41 ± 1.56 | 83 | 0.329 | |
| PMA | D+E space | 15.07 ± 2.79 | 25 | 15.50 ± 1.69 | 23 | 0.528 | | 16.75 ± 1.81 | 46 | 16.88 ± 2.23 | 43 | 0.768 | |
| | C+D+E space | 22.46 ± 3.19 | 25 | 22.05 ± 3.70 | 24 | 0.681 | | 22.88 ± 1.90 | 46 | 22.84 ± 2.17 | 42 | 0.928 | |

**Note:**
NPC, non proximal cavity; PCCP, proximal cavity with buccal or lingual surface contact point to adjacent tooth; PCWC, proximal cavity without contact point to adjacent tooth; PMA, premature loss of primary molar from carious lesions.

pulpal involvement, and five children had tooth number anomalies (supernumerary or missing lateral primary incisor). Moreover, after treatment it was impossible to obtain impressions of the upper arch of two girls because of severe nausea (Fig. 2). In addition, 37 pairs of casts were unavailable for morphometry as they had been either damaged or lost during transportation or because the plaster casts did not match to the respective children. Ultimately, a total of 232 pairs of casts (130 boys and 103 girls) and two lower casts were measured.

The mean age (±SD) of the children was 6.8 (±0.4) years and no differences in sex (maxilla $p = 0.903$, mandible $p = 0.298$) and age (maxilla $p = 0.889$, mandible $p = 0.956$) were observed between groups.

## Error of the method

Inter-examiner agreement on status of primary molars was $\kappa = 0.801$ indicating substantial agreement between examiners. The intra-examiner reliability for C+D+E and D+E spaces is presented in Table 1.

## Primary molar status and intra-arch variables

Table 2 shows the comparison between the intra-arch variables (D+E and C+D+E spaces) for the left and right side for maxilla and mandible separately, according to the primary

**Table 3 C+D+E and D+E spaces (Mean ± SD in mm) in the maxilla and mandible by primary molar status.**

| Variable | | NPC | | | PCCP | | | PCWP | | | PMA | | | P |
|---|---|---|---|---|---|---|---|---|---|---|---|---|---|---|
| | | Mean ± SD | 95% CI | N | Mean ± SD | 95% CI | N | Mean ± SD | 95% CI | N | Mean ± SD | 95% CI | N | |
| Maxilla | D+E space | 16.76 ± 1.02[a] | [16.60–16.91] | 167 | 16.70 ± 0.97[ab] | [16.51–16.90] | 96 | 16.34 ± 1.00[b] | [16.18–16.50] | 150 | 15.28 ± 2.32[c] | [14.61–15.95] | 48 | <0.001 |
| | C+D+E space | 23.89 ± 1.34[a] | [23.69–24.10] | 167 | 24.05 ± 1.34[a] | [23.77–24.32] | 96 | 23.41 ± 1.45[b] | [23.17–23.64] | 150 | 22.57 ± 2.67[c] | [21.80–23.35] | 48 | <0.001 |
| Mandible | D+E space | 18.35 ± 0.99[a] | [18.18–18.53] | 125 | 18.09 ± 0.93[a] | [17.90–18.29] | 89 | 17.51 ± 1.16[b] | [17.33–17.69] | 160 | 16.82 ± 1.99[c] | [16.41–17.24] | 91 | <0.001 |
| | C+D+E space | 24.34 ± 1.38[a] | [24.10–24.58] | 125 | 24.10 ± 1.35[a] | [23.82–24.39] | 88 | 23.52 ± 1.53[b] | [23.28–23.76] | 160 | 22.86 ± 2.00[c] | [22.44–23.28] | 90 | <0.001 |

**Notes:**
NPC, non proximal cavity; PCCP, proximal cavity with buccal or lingual surface contact point to adjacent tooth; PCWC, proximal cavity without contact point to adjacent tooth; PMA, premature loss of primary molar from carious lesions.
Groups with the same letter within the line do not differ significantly.

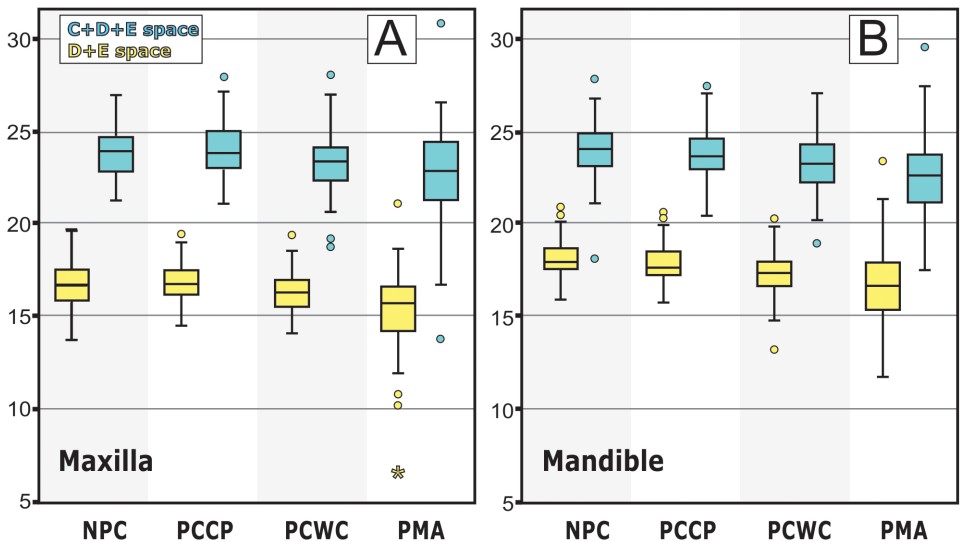

**Figure 3 Graphs show D+E and C+D+E spaces means and standard deviations for the four groups on both arches.** (A) Analyses of Maxilla. (B) Analyses of Mandible. NPC, non proximal cavity; PCCP, proximal cavity with buccal or lingual surface contact point to adjacent tooth; PCWC, proximal cavity without contact point to adjacent tooth; PMA, premature loss of primary molar from carious lesions.

molar status. No statistically significant difference was found between the sides for upper and lower D+E and C+D+E spaces. This implied that both sides were pooled to compare the association of the intra-arch variables and "status of primary molar".

ANOVAs and post hoc Tukey's tests were applied to assess the impact of the primary molar's status (NPC, PCCP, PCWC and PMA) on the dependent variables (C+D+E space and D+E space) (Table 3). The results showed statistically significant differences ($P < 0.001$) between the independent and dependent variables in both arches. The quadrants that showed absence of primary molars (PMA group) had the smallest intra-arch spaces (D+E and C+D+E spaces), with a gradient increase towards PCWC, PCCP and NPC groups (Fig. 3).

## DISCUSSION

The null-hypothesis was rejected as we found some differences in the C+D+E and D+E spaces of schoolchildren in relation to the status of the proximal surface of posterior primary molars.

Data from this deprived community provided a unique source of ethically collected information to investigate whether proximal cavitated dentin carious lesions and premature loss of primary molars affect D+E and C+D+E spaces, since it is unacceptable to leave children with dentin carious lesions in the primary molars without dental treatment. Thus, the UCT treatment protocol provided an invaluable model for assessing the influence of cavities and tooth loss in the primary and mixed dentition on malocclusion.

Our sample was divided into four groups with the primary aim to elucidate how different levels of proximal caries severity—taking tooth loss as the final negative effect of dental caries—influence the C+D+E and D+E spaces. An interesting observation was that partial contact points as presented in the PCCP group, while obviously not the normal anatomical contact points of sound or conventionally restored teeth, appeared to be sufficient to keep space in the arch. This explains why the NPC and the PCCP did not differ from each other (Table 3).

On the other hand, proximal cavities without contact point to adjacent teeth (PCWC group) led to space loss in both arches, and additional space loss took place in our sample when primary molars were absent.

Several studies in the past focused on the possible association between the DMFT index and occlusal traits to investigate the association of caries and malocclusion (*Stahl & Grabowski, 2004*; *Gábris, Márton & Madlénaa, 2006*; *Mtaya, Brudvik & Astrøm, 2009*; *Borzabadi-Farahani, Eslamipour & Asgari, 2011*; *Nalcaci et al., 2012*). However, this index might not be an adequate tool to approach the issue since, as our results suggest, the subtle levels of caries severity of the proximal surface resulting in contact or no contact point with the neighboring tooth do impact on mesio-distal tooth migration. The DMFT index, however, does not capture the contact status between adjacent teeth.

Our results slightly differ from what was observed in a 6-year longitudinal study that found that large cavities tended to decrease the D+E space (*Northway & Wainright, 1980*). It is likely that for most individuals the amount of space loss, depicted in our study, might not be of clinical relevance. It is also worth noting that it represents less than the leeway space. However, the clinical relevance may need to be analyzed on an individual basis. A near half-millimetre space loss taking place when a contact point with an adjacent tooth is missing or a less-than-one-millimetre space loss when the primary molar is absent may be a problem for those individuals who already lack space for future proper alignment of the permanent dentition, particularly if the condition affects both sides.

In fact, both dependent variables showed similar performances, suggesting that primary canines are affected by primary molar proximal cavities or early tooth loss at the same amount as the primary molars are affected. In other words, when the primary molars drift distally, they are followed by the primary canines.

Although the use of standardized photographs and plaster models may appear outdated in the digital era, our methodology had been affirmed before. The C+D+E and D+E spaces are linear distances which makes two-dimensional tools sufficient for measuring them. Also, the standardized photographs enable measurements to be performed from the same angle of view with an enlargement between 2 and 10 times, which is not possible when measuring directly at the plaster model.

The findings of the present study have orthodontic consequences as well as implications for restorative treatment in the primary and mixed dentition. When orthodontic treatment is needed anyway and space losses of below leeway space dimensions will not affect the outcome, no space maintainers or CRT (e.g. amalgam) is necessary. Herein are most of the extraction cases, temporary skeletal anchorage cases (when the temporary skeletal anchorage might be used as anchorage unit to gain space), and others. Nonetheless, this decision must be made on an individual basis. By the same token, if minor space loss may not be clinically relevant, the ultra-conservative treatment is a viable and sound treatment option.

It is important to mention that this is a cross-sectional study, with the known limitations of this study design. However, a longitudinal study is being prepared and will be published soon. The current study had some additional limitations. First, we could not differentiate the extractions that took place just before impression taking from those which had been performed much earlier. Thus, we could not quantify space loss over time. Also, we did not stratify our sample to identify possible contrasting influences that first or second primary molar losses may have. Stratification would have led to inflation of type I error rate and groups with a small sample size. Second, some confounding variables—such as permanent first molar eruption status—were not considered. During permanent first molar eruption, space loss may occur. The American Academy of Pediatric Dentistry supports the insertion of space maintainers to replace first primary molars when permanent first molars are erupting (*American Academy of Pediatric Dentistry, 2019*). Also, it has been suggested that children with end-to-end molar occlusion and hyperdivergent facial biotype may be more susceptible to tooth migration (*Alexander, Askari & Lewis, 2015*).

## CONCLUSION

Proximal cavities in primary molars without contact point to adjacent teeth are associated with minor C+D+E and D+E space loss in both dental arches. Additional space loss is noted when primary molars are absent. In this study, no space loss took place when buccal or lingual surfaces still provided contact points with adjacent teeth. These results may have implications for orthodontic and restorative dental care decisions in children.

## ACKNOWLEDGEMENTS

The authors thank children, parents, principals and teachers of the public schools in Paranoá for participating in the study. Special thanks are extended to Rodrigo Ferreira Guedes de Amorim for providing relevant information from the primary study, Simone

Moraes Otero for taking impressions, and all dentists and assistants who participated. We are very thankful to Dr. E. Bronkhorst for analysing the data.

### Funding
This work was supported by the Federal District Foundation (FAP-DF) of Brazil, number 193.000.381/2008, and by the Radboud University Medical Centre, Nijmegen, the Netherlands, number R00001285. The funders had no role in study design, data collection and analysis, decision to publish, or preparation of the manuscript.

### Grant Disclosures
The following grant information was disclosed by the authors:
Federal District Foundation (FAP-DF): 193.000.381/2008.
Radboud University Medical Centre, Nijmegen, the Netherlands: R00001285.

### Competing Interests
Anne Marie Kuijpers-Jagtman is an Academic Editor for PeerJ.
 All the remaining authors declare that they have no competing interests.

### Author Contributions
- Rafael T. Gomide conceived and designed the experiments, performed the experiments, prepared figures and/or tables, authored or reviewed drafts of the paper, and approved the final draft.
- Jo E. Frencken authored or reviewed drafts of the paper, and approved the final draft.
- Soraya C. Leal conceived and designed the experiments, authored or reviewed drafts of the paper, and approved the final draft.
- Anne Marie Kuijpers-Jagtman authored or reviewed drafts of the paper, and approved the final draft.
- Jorge Faber conceived and designed the experiments, prepared figures and/or tables, authored or reviewed drafts of the paper, and approved the final draft.

### Human Ethics
The following information was supplied relating to ethical approvals (i.e., approving body and any reference numbers):
 This study was approved by the Research Ethics Committee of the University of Brasília Medical School (protocol 081/2008).

### Data Availability
 The raw measurements are available in the Supplemental Files.

### Supplemental Information
Supplemental information for this article can be found online at http://dx.doi.org/10.7717/peerj.8924#supplemental-information.

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
