# Peer review of "Impact of proximal cavities and primary molar absence on space in the dental arches"

_PeerJ, doi:10.7717/peerj.8924_

## Round 0.1 · original submission · Major Revisions

· Academic Editor

Major Revisions

This study is important and adds important information to the existing literature that can impact on advocating for the prompt clinical management of carious primary teeth. There are however a few flaws that need to be addressed
1. The English needs to be improved. It will be important for an English editor to read the manuscript and help improve the quality
2. The writer needs to explain what these measures are of importance and interest in the introduction - C+D+E and D+E. Why did they use two parameters?
3. The source of the data for this cross-sectional analysis needs further clarity. The authors refer to a clinical trial and an epidemiological survey. if there is a link please make this clearer as the source of this data is not clear
4. Since this study only used the baseline casts and data for these children, I suggest the details about the clinical trial be reduced to just two paragraphs at the most as it creates confusion for the readers. It may be more important to highlight details of the baseline measures and making of the cast for this study
5. Line 163 – were the cast used for this study for only children who had caries restoration. Where there no children who had their cast taken even though they had no caries lesion or had caries and did not have to have a restoration?
6. Line 177 - Important to report the result of the calibration
7. Line 196 – it is difficult to verify the authencity of this deduction in the absence of information on the age range of the study participants at baseline
8. The casts were taken AFTER the restorations
9. With whom was the inter-examiner reliability done for this single examiner of the cast?
10. I suggest the authors improve the writing of the methodology section using the STROBE guidance. The current outline is difficult to follow. Also the data analysis plan is not clear – please ensure the explanatory and outcome variables are well defined. For example, dmft is mentioned as an explanatory variable but its measure was not discussed. There are also a number of other variables that showed up in the data analysis plan that there was no clarity about it extraction from the data set in the earlier method details
11. Line 227 -The results say 302 children were enrolled. A prior statement had referred to the N of this study as 835. Similarly, a prior detail on the clinical trial had reported on the N as 80 children. This is all not clear at this point. The data on the subjects should be in the methodology as this is a secondary data analysis. Clarity about the sample size is a methodology detail and not a result detail
12. Line 236 – looks like the means reported here is for sex and age related dmft. This is not clear
13. Please just focus the result section on the results of the study and please report what the statistically analysis implied.
14. Line 247 – difficult to understand the statement - C+D+E spaces were affected by ‘status of primary molar’ (P<0.001) in both arches (Table 3).
15. Please focus discussion on study finding. The discussion is currently too long for the findings. Please use the STOBE guidelines for reporting cross-sectional study

Reviewer 1 ·

Basic reporting

Use of English language is clear and unambiguous, but authors should kindly rephrase line 33 and 34 in the abstract session.

The study title is clear and reflects the content in the proposal. The title, aims and objectives form a meaningful whole

The concepts are well defined, and the study is well conceptualized

The literature review is relevant adequate and recent.

Experimental design

Authors should kindly indicate why the choice of space analysis using photographs of casts rather than direct measurement/assessment of space from the casts.

Validity of the findings

• Authors should kindly include in limitation session that radiographic assessment/analysis of space was not done

Reviewer 2 ·

Basic reporting

Improvement is required in overall writing as the text is not aptly able to convey the message.

Experimental design

The authors have done a cross-sectional study - which seems to be done in continuation to already done a clinical trial. But clear explanation of the same is somewhat obscure.

Validity of the findings

It seems OK.

Additional comments

Overall clarity is missing in the manuscript. It seems you have done a cross-sectional study - which seems to be done in continuation to already done a clinical trial. But clear explanation of the same is somewhat obscure.

---

## Round 0.2 · Major Revisions

· Academic Editor

Major Revisions

These suggested edits were not addressed. I did not find the response to these suggested edits in the rebuttal letter

This study is important and adds important information to the existing literature that can impact on advocating for the prompt clinical management of carious primary teeth. There are however a few flaws that need to be addressed
1. The English needs to be improved. It will be important for an English editor to read the manuscript and help improve the quality
2. The writer needs to explain what these measures are of importance and interest in the introduction - C+D+E and D+E. Why did they use two parameters?
3. The source of the data for this cross-sectional analysis needs further clarity. The authors refer to a clinical trial and an epidemiological survey. if there is a link please make this clearer as the source of this data is not clear
4. Since this study only used the baseline casts and data for these children, I suggest the details about the clinical trial be reduced to just two paragraphs at the most as it creates confusion for the readers. It may be more important to highlight details of the baseline measures and making of the cast for this study
5. Line 163 – were the cast used for this study for only children who had caries restoration. Where there no children who had their cast taken even though they had no caries lesion or had caries and did not have to have a restoration?
6. Line 177 - Important to report the result of the calibration
7. Line 196 – it is difficult to verify the authenticity of this deduction in the absence of information on the age range of the study participants at baseline
8. The casts were taken AFTER the restorations
9. With whom was the inter-examiner reliability done for this single examiner of the cast?
10. I suggest the authors improve the writing of the methodology section using the STROBE guidance. The current outline is difficult to follow. Also the data analysis plan is not clear – please ensure the explanatory and outcome variables are well defined. For example, dmft is mentioned as an explanatory variable but its measure was not discussed. There are also a number of other variables that showed up in the data analysis plan that there was no clarity about it extraction from the data set in the earlier method details
11. Line 227 -The results say 302 children were enrolled. A prior statement had referred to the N of this study as 835. Similarly, a prior detail on the clinical trial had reported on the N as 80 children. This is all not clear at this point. The data on the subjects should be in the methodology as this is a secondary data analysis. Clarity about the sample size is a methodology detail and not a result detail
12. Line 236 – looks like the means reported here is for sex and age related dmft. This is not clear
13. Please just focus the result section on the results of the study and please report what the statistical analysis implied.
14. Line 247 – difficult to understand the statement - C+D+E spaces were affected by ‘status of primary molar’ (P<0.001) in both arches (Table 3).
15. Please focus discussion on study findings. The discussion is currently too long for the findings. Please use the STROBE guidelines for reporting cross-sectional study

Also, please use a consistent terminology - primary teeth not deciduous teeth.

---

## Round 0.3 · accepted · Accept

· Academic Editor

Accept

The manuscript is really improved and easier to read and follow. Glad the authors took time to address the concerns raised. Wishing you well with a manuscript that is important for the field.